# Selective sweeps on different pigmentation genes mediate convergent evolution of island melanism in two incipient bird species

Leonardo Campagna[1,2]*, Ziyi Mo[3,4], Adam Siepel[3], J. Albert C. Uy[5]*

1 Fuller Evolutionary Biology Program, Cornell Lab of Ornithology, Ithaca, New York, United States of America, 2 Department of Ecology and Evolutionary Biology, Cornell University, Ithaca, New York, United States of America, 3 Simons Center for Quantitative Biology, Cold Spring Harbor Laboratory, Cold Spring Harbor, New York, United States of America, 4 School of Biological Sciences, Cold Spring Harbor Laboratory, Cold Spring Harbor, New York, United States of America, 5 Department of Biology, University of Rochester, Rochester, New York, United States of America

* lc736@cornell.edu (LC); al.uy@rochester.edu (JACU)

**Data Availability Statement:** The computer code for this project has been deposited in GitHub, https://github.com/CshlSiepelLab/bird_capuchino_analysis and https://github.com/CshlSiepelLab/arg-

## Abstract

Insular organisms often evolve predictable phenotypes, like flightlessness, extreme body sizes, or increased melanin deposition. The evolutionary forces and molecular targets mediating these patterns remain mostly unknown. Here we study the Chestnut-bellied Monarch (*Monarcha castaneiventris*) from the Solomon Islands, a complex of closely related subspecies in the early stages of speciation. On the large island of Makira *M. c. megarhynchus* has a chestnut belly, whereas on the small satellite islands of Ugi, and Santa Ana and Santa Catalina (SA/SC) *M. c. ugiensis* is entirely iridescent blue-black (i.e., melanic). Melanism has likely evolved twice, as the Ugi and SA/SC populations were established independently. To investigate the genetic basis of melanism on each island we generated whole genome sequence data from all three populations. Non-synonymous mutations at the *MC1R* pigmentation gene are associated with melanism on SA/SC, while *ASIP*, an antagonistic ligand of *MC1R*, is associated with melanism on Ugi. Both genes show evidence of selective sweeps in traditional summary statistics and statistics derived from the ancestral recombination graph (ARG). Using the ARG in combination with machine learning, we inferred selection strength, timing of onset and allele frequency trajectories. *MC1R* shows evidence of a recent, strong, soft selective sweep. The region including *ASIP* shows more complex signatures; however, we find evidence for sweeps in mutations near *ASIP*, which are comparatively older than those on *MC1R* and have been under relatively strong selection. Overall, our study shows convergent melanism results from selective sweeps at independent molecular targets, evolving in taxa where coloration likely mediates reproductive isolation with the neighboring chestnut-bellied subspecies.

## Author summary

Chestnut-bellied Monarchs (*Monarcha castaneiventris ugiensis*) from two archipelagos in the Solomon Islands have evolved entirely black plumage from a chestnut ancestor

selection. Genomic data have been archived in GenBank (BioProject ID PRJNA835722).

**Funding:** A National Science Foundation CAREER Award (IOS 1137624/0643606; https://beta.nsf. gov/) and a National Geographic Society CRE Award (9023-11; www.nationalgeographic.org/) to JACU supported field work to collect samples. The School of Arts & Sciences at the University of Rochester (https://www.sas.rochester.edu/) funded the acquisition of sequence data (funds obtained by JACU). The funders had no role in study design, data collection and analysis, decision to publish, or preparation of the manuscript.

**Competing interests:** The authors have declared that no competing interests exist.

(*Monarcha castaneiventris megarhynchus*), a phenomenon known as island melanism. We obtain and analyze whole genome sequences using traditional summary statistics and new methods that combine inference of the ancestral recombination graph with machine learning. We find multiple lines of evidence for independent selective sweeps on the *MC1R* and *ASIP* genes, a receptor/ligand pair which regulates the production of melanin. Melanism on each archipelago is mediated by mutations in one of these two genes. Mutations in and around *MC1R* underwent a recent soft sweep experiencing strong selection on the islands of Santa Ana and Santa Catalina, whereas selection was also strong but comparatively older for *ASIP* on the island of Ugi. We show how melanism originated under positive selection on independent molecular targets, evolving convergently in taxa where coloration mediates reproductive isolation.

## Introduction

The extent to which evolutionary change can be predicted has been a longstanding matter of debate in evolutionary biology [1–3]. Instances of convergent evolution support the argument that evolutionary change can be deterministic, yet stochastic historical events can lead to divergent outcomes from recently split taxa. A better understanding of the eco-evolutionary forces and genetic mechanisms behind evolutionary changes will shed light on the conditions under which deterministic or stochastic outcomes can occur. Some examples of convergent evolution occurred deep in the tree of life, like the independent origins of wings in birds, bats and insects [2], while other cases represent more recent (and potentially ongoing) phenomena like the repeated radiations of ecomorphs in Caribbean lizards [4], the loss of flight associated to insularity in insects and birds [5,6] or the evolution of island melanism [7]. These recent classic examples of phenotypic convergence can be leveraged to study the evolutionary forces and molecular mechanisms behind phenotypic change. Here we focus on island melanism in birds, a phenotype that involves the increased deposition of eumelanin, which leads to entirely black plumage coloration [8–10].

The Chestnut-bellied Monarch (*Monarcha castaneiventris*) from the Solomon Islands represents a complex of closely related subspecies which are in the early stages of speciation and vary in plumage color, song, and body size [11–13]. One of these subspecies, *M. c. ugiensis*, has entirely iridescent blue-black plumage, and is found on the small satellite islands to the north and southeast of the larger island of Makira (Fig 1A). In contrast, the endemic subspecies on Makira is *M. c. megarhynchus* and has a chestnut belly and iridescent blue-black upper parts. Phylogenetic analyses using reduced-representation genomic data show that *M. c. ugiensis* individuals from the satellite islands of Ugi, and Santa Ana and Santa Catalina (SA/SC) are independently derived from the chestnut-bellied Makira population, suggesting that *M. c. ugiensis* is polyphyletic and melanism has evolved repeatedly and convergently [14]. A candidate gene study suggested that the molecular basis of increased melanin deposition differs between the Ugi and SA/SC populations [9]. Melanism on each of the satellite islands is associated with mutations that affect the coding sequence of the *MC1R/ASIP* receptor and ligand pair, two molecules that regulate the balance between the production of eumelanin (a pigment conferring black/gray coloration) and pheomelanin (a pigment which leads to brown/yellow coloration). While the melanic individuals from SA/SC carry a derived non-synonymous mutation on the *MC1R* receptor, their counterparts from Ugi possess a non-synonymous mutation on the *ASIP* ligand, and heterozygotes at either mutation display an intermediate coloration phenotype (Fig 1B; [9]). Finally, it is likely that changes in plumage color mediated

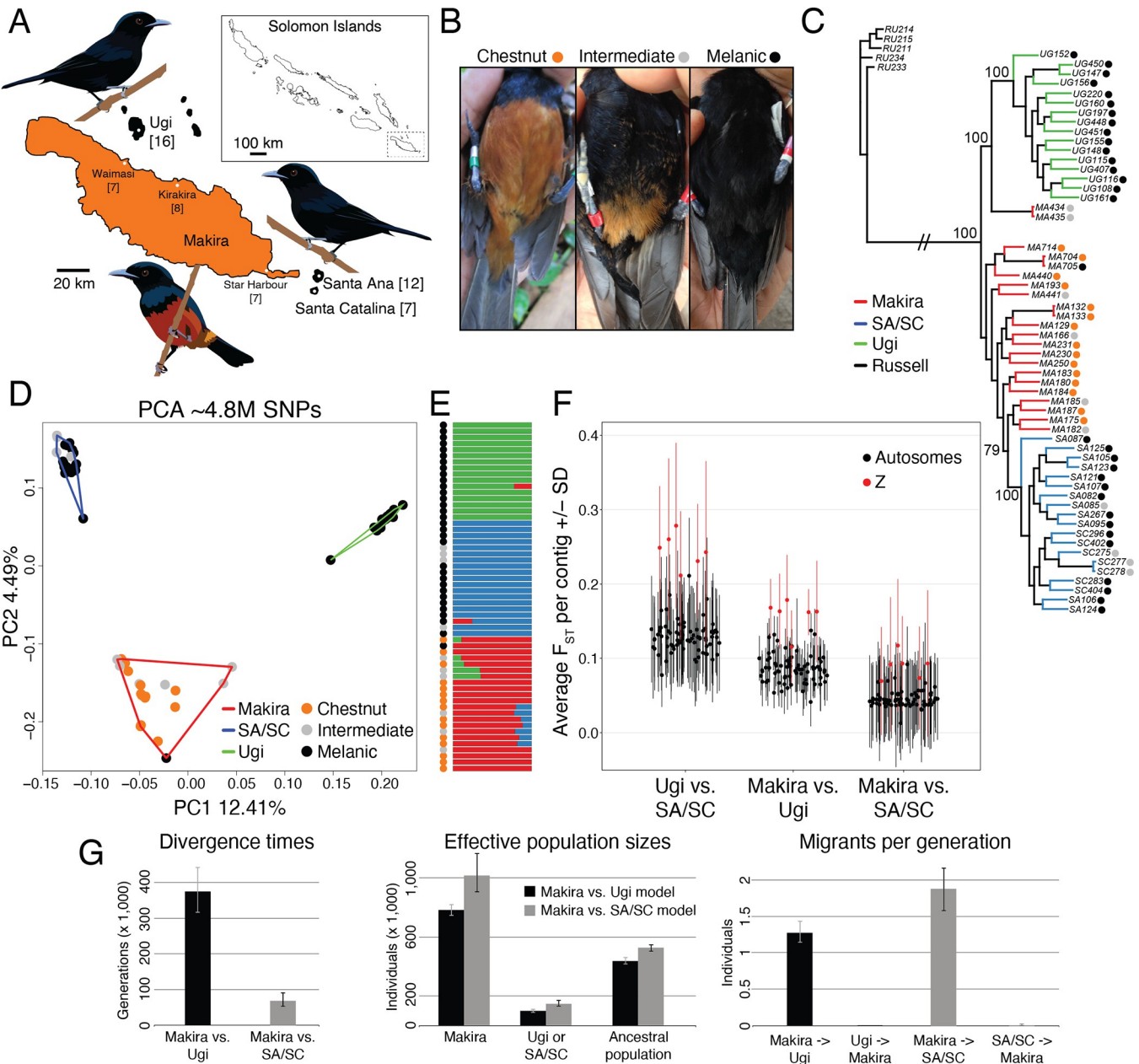

**Fig 1. Genetic differentiation and demography of *M. c. megarhynchus* and *M. c. ugiensis*. A.** Study area, sample sizes and predominant phenotype on each island. The map was downloaded and modified from www.diva-gis.org. **B.** Representative pictures of chestnut-bellied, intermediate and melanic individuals (color key used throughout the manuscript). Maximum Likelihood tree (**C**) and PCA (**D**) indicating the origin and coloration phenotype of each individual. **E.** Admixture plot showing the proportion of ancestry for each individual belonging to three different genetic clusters. Each cluster is color-coded by the island from which samples originated and the phenotype is shown by color-coded circles on the left of the plot. **F.** Pairwise $F_{ST}$ estimates summarized by contig. **G.** Demographic reconstructions indicating estimates of divergence times, effective population sizes, and migrants per generation.

by these mutations generate prezygotic reproductive isolation between the melanic populations on the satellite islands and the chestnut-bellied population on nearby Makira, as territorial males discriminate individuals by their phenotype, and respond predominantly to simulated territorial intrusions of males with the local plumage (and song) traits [15,16]. Convergent melanism, therefore, may result in repeated speciation between the chestnut-bellied population of Makira, and each of the two melanic populations of Ugi and SA/SC.

Here we generate a reference genome for the Chestnut-bellied Monarch and obtain high coverage whole-genome data for a sample of individuals from Makira and its satellite islands. Our study aims to uncover the molecular targets and evolutionary forces that shape convergent evolution of adaptive traits that can contribute to generating prezygotic reproductive isolation. We use these data to quantify differentiation, reconstruct phylogenetic affinities, and infer the demographic history of these populations. We then use a genome-wide approach to identify variants associated with melanic plumage. Finally, we infer the evolutionary processes that have shaped these phenotypes on each of the satellite islands of Ugi and SA/SC, estimate when mutations arose and the timing of these selective events.

## Results

### Melanic populations are independently derived from a chestnut-bellied ancestor

Birds grouped together by island, irrespective of their coloration phenotype (Fig 1C and 1D). Individuals from the satellite islands of SA/SC (which are primarily melanic, yet show a low prevalence of the intermediate coloration phenotype) and Ugi (which are exclusively melanic) formed island-specific clades which were embedded among clades containing primarily chestnut-bellied individuals from the larger island of Makira (Fig 1C). The relationships among birds from the three islands could not be resolved using mtDNA, as individuals from every locality share haplotypes (S1 Fig). Consequently, melanism on the two satellite islands likely originated twice, independently from a chestnut-bellied ancestor [9,14]. We did not observe clear evidence of early generation inter-island hybrids in the genome-wide PCA (Fig 1D), however the two individuals from Makira which form a clade with the individuals from Ugi (Fig 1C) showed intermediate coloration and were sampled in the locality which is closest to Ugi (Waimasi), suggesting the possibility of either incomplete lineage sorting or gene flow. Furthermore, we observed Makira ancestry in one individual of each of the satellite islands, and SA/SC or Ugi ancestry in a few individuals on Makira (Fig 1E). The admixed individuals on Makira were from the localities closest to the satellite island with which they shared ancestry (Waimasi for Ugi and Star Harbour for SA/SC). The levels of differentiation among populations were largest between Ugi and SA/SC, intermediate between Ugi and Makira, and smallest between SA/SC and Makira (Fig 1F). The contigs showing the highest differentiation for each pairwise population comparison were in all cases Z-linked. The difference in the magnitude of genetic differentiation between populations could be due to variation in a combination of demographic parameters (i.e, the splitting time, the degree of gene flow experienced after this split, or the intensity of genetic drift due to differences in effective population sizes). We therefore used sequence data to conduct a demographic reconstruction with G-PhoCS, which suggested that the main reason for the observed difference in the levels of differentiation between populations was that Ugi split from Makira approximately six times earlier than SA/SC branched from Makira (Fig 1G). Additionally, the effect of genetic drift is likely to be slightly stronger in Ugi, as its effective population size was inferred to be nearly two thirds of that of SA/SC (and approximately one eighth of Makira's). Finally, G-PhoCS inferred significant levels of gene flow from Makira into each of the satellite islands (higher into SA/SC) and not in the reverse direction (Fig 1G), suggesting that the admixture observed on Makira (Fig 1E) may be due to the retention of ancestral polymorphisms in this larger population.

### Melanism on each satellite island associates with mutations in different genes

To test if the convergent melanic phenotype on each of the satellite islands was also convergent at the molecular level, we conducted two genome wide association studies while controlling

for population structure by including an inter-individual relatedness matrix as a covariate. The first included individuals from Makira and SA/SC and revealed a single peak on contig 400 (corresponding to chromosome 11) composed of 61 SNPs with association values above the significance threshold (Fig 2A). This region contained 15 annotated genes, including the coloration gene *MC1R* (Fig 2B and S1 Table). The second GWAS, derived from individuals from Ugi and Makira showed seven association peaks with 83 annotated genes (Fig 2C and S1 Table), suggesting a larger number of genes could mediate melanism on Ugi. One of these association peaks, on contig 947 (located on chromosome 20), contained four of the six strongest hits in the GWAS. The *MC1R* antagonist *ASIP* was one of the 14 genes in this region (Fig 2D). The variants within the seven association peaks were in high linkage disequilibrium (average intrachromosomal R2 = 0.84; average interchromosomal R2 = 0.79; S2 Fig). We did not find other known coloration genes within the remaining association peaks (S1 Table), suggesting these genes have unknown functions in melanism or mediate other differences between the Ugi and Makira populations which may covary with changes in coloration (e.g., Ugi individuals are larger than those from Makira). Three of the seven association peaks were on the Z sex chromosome, which is consistent with this chromosome evolving faster than the autosomes in birds [17]. Additionally, when comparing within the region encompassed by the association peaks containing *MC1R* and *ASIP* and outside of this region (for contig 400 and 947 separately), we observed higher levels of differentiation between Makira and each of the satellite islands (S3A and S3B Fig).

The melanic individuals from SA/SC had two haplotypes in the region which contained the 61 association hits on contig 400, which were different from the most prevalent haplotype on Makira and Ugi (S4 and S5 Figs). Three variants fell within the coding region of *MC1R*; two of these positions involved synonymous changes and one coded for an *Asp119Asn* substitution. Similarly, all the individuals from Ugi possessed two haplotypes that were different from the main one present in SA/SC and Makira individuals in the association region around *ASIP* (38 SNPs; S5 and S6 Figs). A single position fell within the coding region of *ASIP* and involved a non-synonymous *Ile55Thr* substitution, with all Ugi individuals carrying the *Thr55* allele. In conclusion, melanic individuals always carried two copies of the coding *MC1R* mutation (*Asn*119) observed on SA/SC or of the coding *ASIP* mutation (*Thr*55) observed on Ugi.

## The regions of the genome containing *MC1R* and *ASIP* show signatures of selective sweeps

We first searched for signatures of selection by calculating summary statistics from the focal contigs containing coloration genes. The region which includes the *MC1R* gene produced negative values of Tajima's D and low nucleotide diversity in the SA/SC population (Fig 3A), as expected for a selective sweep, and high H12 and intermediate H2/H1 values, which are consistent with a relatively soft selective sweep (Fig 3B). However, because of the windowed nature of this analysis we are cautious in interpreting the specific type of sweep that affected the *MC1R* gene. We observed windows within the peak on contig 947 for the Ugi population that showed an overall similar pattern to the one seen for *MC1R* on SA/SC (Fig 3C and 3D). The positive value of Tajima's D for the window containing *ASIP* on contig 947 (1.4) may be consistent with balancing selection, yet represents an average for a 5 kb window which only included a single SNP (out of 12) from the gene region. In fact, when we calculate Tajima's D for 500 bp windows, the one which includes *ASIP* has a value close to zero (0.25; calculated from 3 SNPs in that window). We opted to present our results for 5 kb windows as these contain an average of 25 SNPs per window (vs. an average of 3 SNPs for 500 bp windows) and therefore represent more robust values of the summary statistics. Finally, we note that

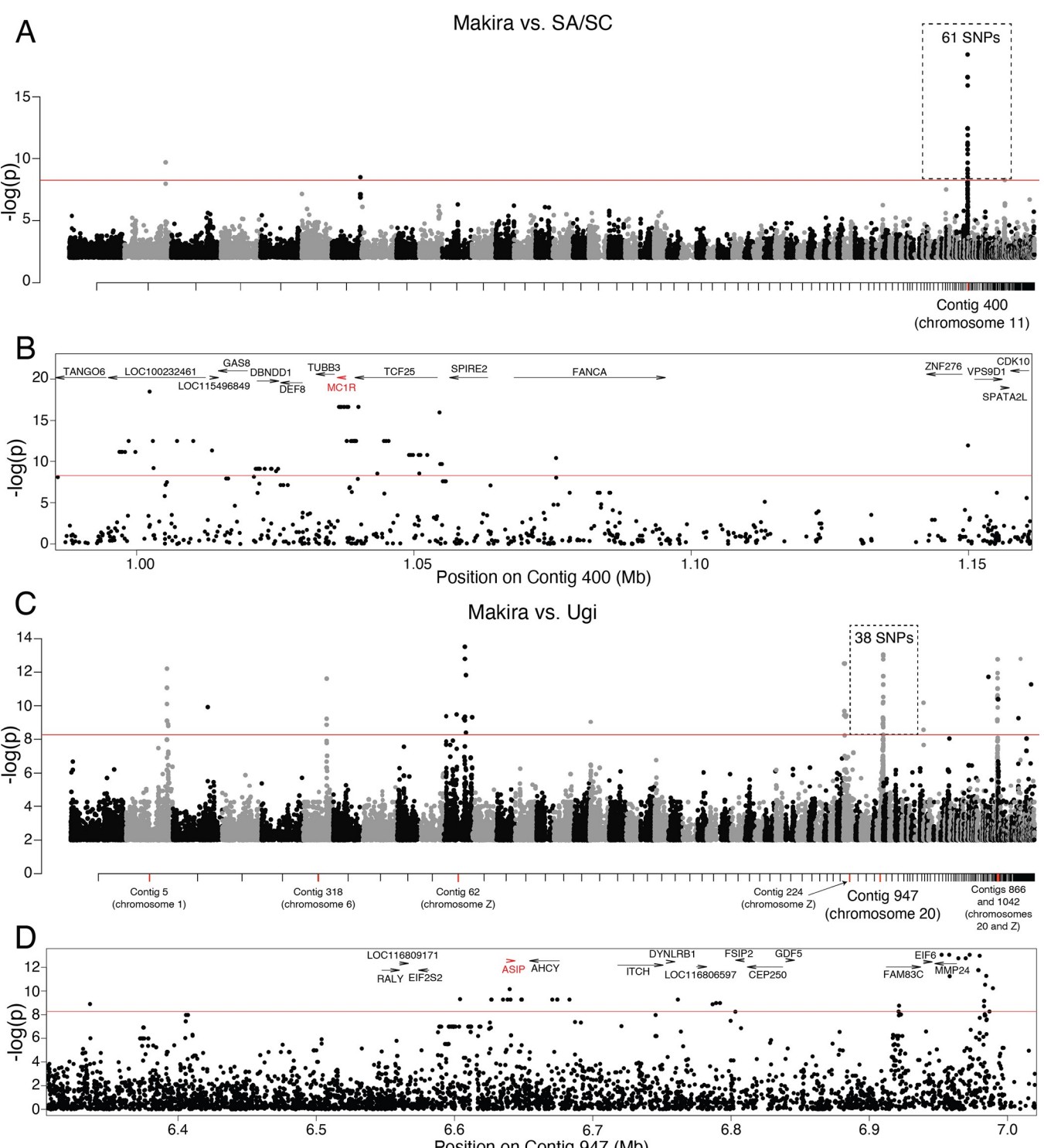

**Fig 2. Genome Wide Association Study. A**. Manhattan plot obtained from the GWAS comparing individuals from Makira and SA/SC. **B**. Zoom-in to the association peak in A indicating gene annotations within this region with *MC1R* in red. Equivalent plots for the GWAS obtained with individuals from Makira and Ugi (**C**, **D**).

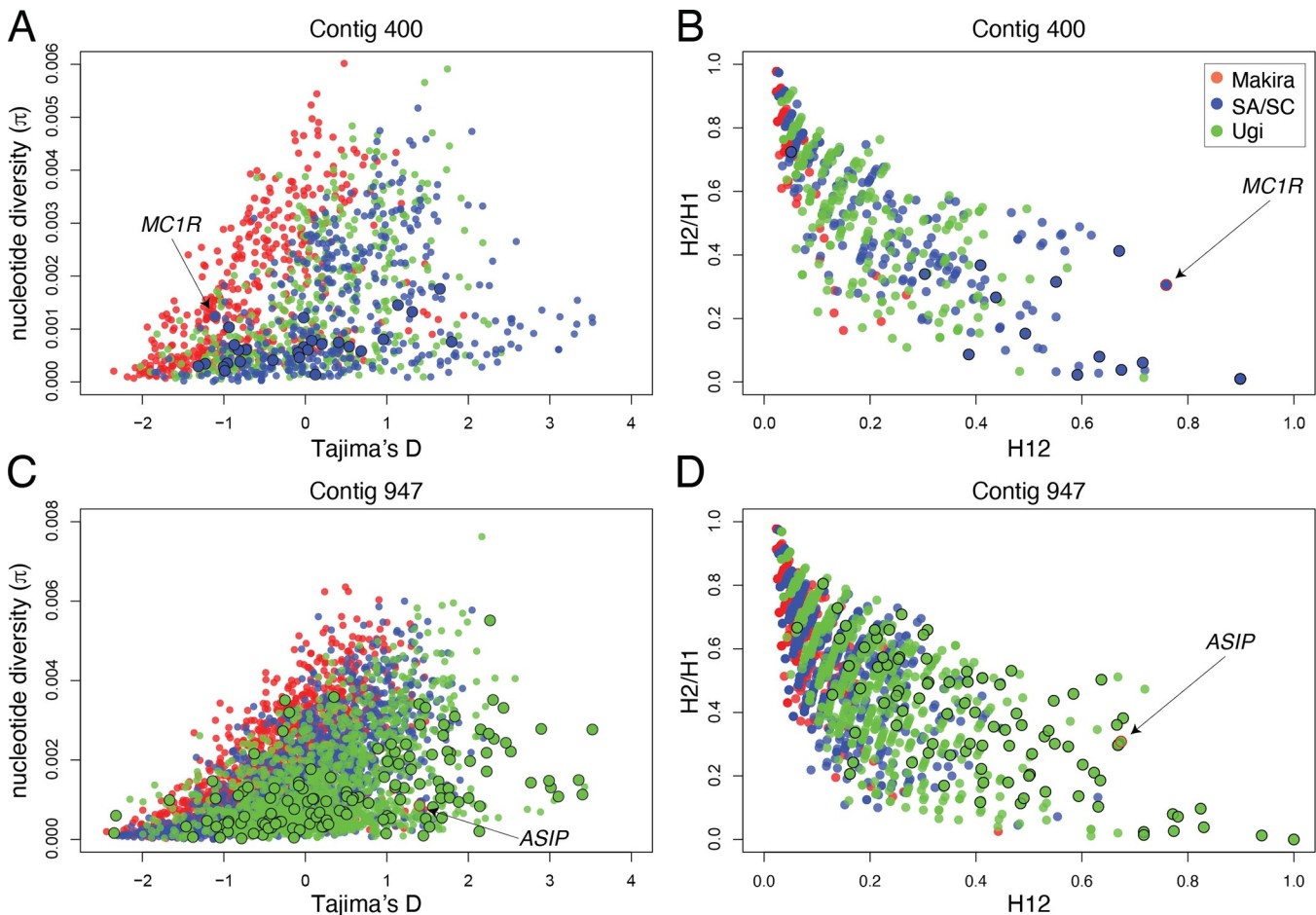

**Fig 3. Evidence of a selective sweep in *MC1R* for the SA/SC population and on *ASIP* in the Ugi population.** Biplots of Tajima's D vs. nucleotide diversity for contig 400 and contig 947 (**A, C**). Biplots of H12 vs. H2/H1 for the same contigs as above (**B, D**). Dots represent statistics derived from 5 kb windows (Tajima's D vs. nucleotide diversity) or 100-SNP windows (H12 vs. H2/H1), and are color coded based on the population of origin. Larger dots denote windows that belong to the outlier peak region, and those that have a red outline include the focal gene indicated by the arrow.

genome-wide values of Tajima's D tend to be close to zero for the three populations (-0.6 for Makira and 0.1 for both SASC and Ugi), which suggests this statistic hasn't been strongly impacted by demographic trends.

We next searched for signatures of selection on the focal contigs by calculating two statistics derived from the ancestral recombination graph (ARG): a species (or population) enrichment score and a measure of normalized time to most recent common ancestry (TMRCA) called the relative TMRCA half-life (RTH'; [18,19], S7A Fig). Species enrichment scores measure the probability of observing subtrees of different sizes containing individuals from a certain species. RTH' is the TMRCA of half of the haploid samples of a species divided by the age of the youngest subtree containing half of all the haploid samples, and measures the age of coalescence events independently of the overall coalescence rate. We reasoned that areas of the genome that have undergone a selective sweep in a given population should show shallow genealogies (low RTH' values) comprising most individuals of that population (high species enrichment score) [19]. We averaged these statistics across 20 kb windows and for each statistic we established population-specific thresholds based on the distribution of values obtained from control windows. In the *MC1R* region of contig 400, we observed a statistically significant

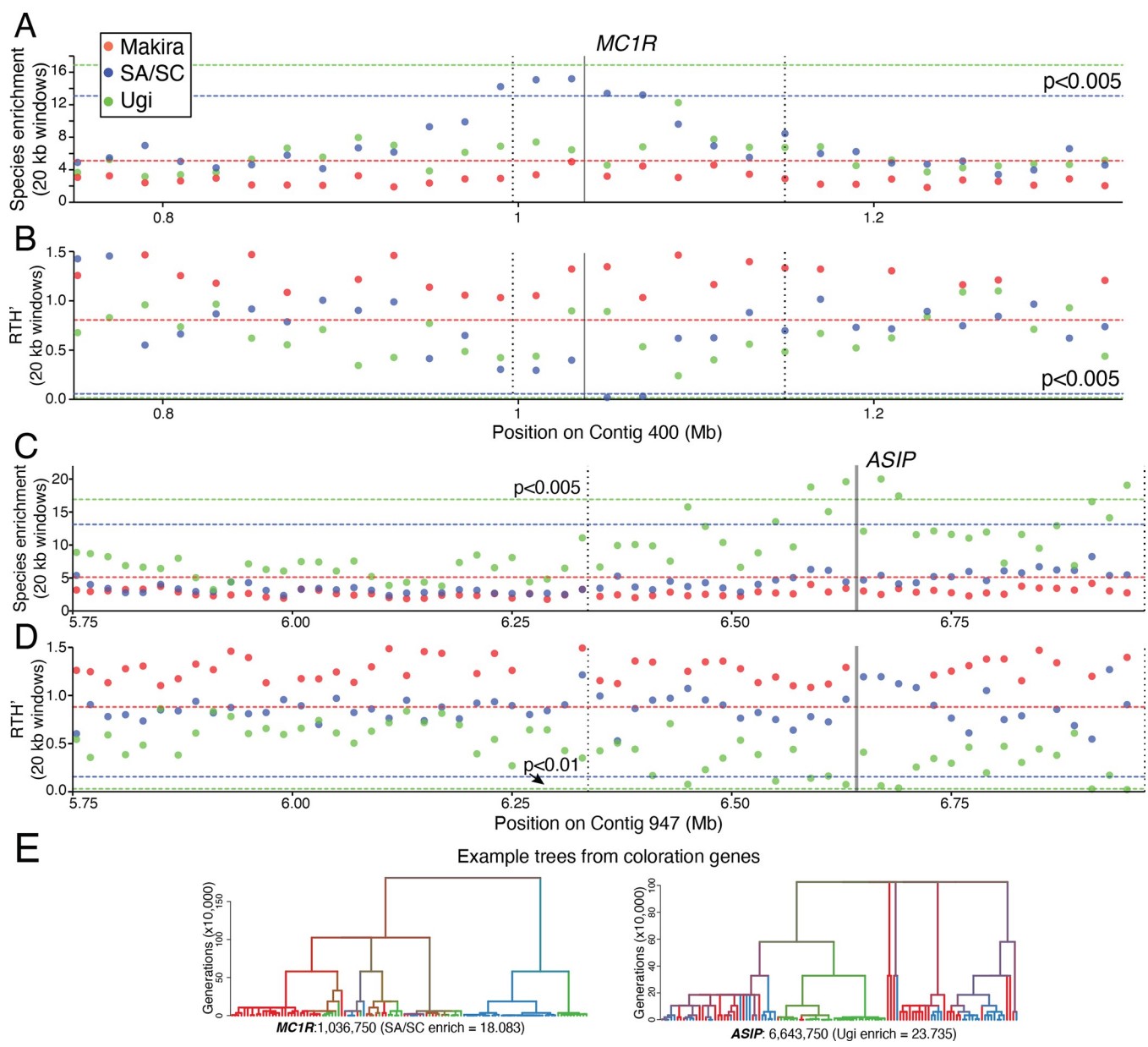

**Fig 4. Signatures of selective sweeps in ARG-based statistics on the focal contigs with coloration genes.** Plots showing species enrichment (**A**, **C**) and RTH' (**B**, **D**) in 20 kb windows along contig 400 and 947. Horizontal lines show species-specific levels of statistical significance, dashed vertical lines define the regions of the association peaks, and solid vertical lines show the position of coloration genes (*MC1R* and *ASIP*). **E**. Outlier values of species-enrichment within the *MC1R* and *ASIP* genes (the position on the contig from which each topology is derived is shown on the bottom of each tree). The terminal branches are color-coded by island and the color of internal branches represents an average over all offspring branches.

elevation of the SA/SC enrichment score, which coincided with a dip in RTH' ($p < 0.005$ in both cases; Fig 4A and 4B). In the *ASIP* region of contig 947, we observed a similar pattern for the Ugi population (enrichment: $p < 0.005$, RTH' $p < 0.01$; Fig 4C and 4D). These statistical outliers were generated from trees with large and shallow population-specific clades (S8 Fig). The statistics for the remaining populations in each of the focal contigs did not surpass the thresholds of statistical significance and resembled the values observed for the control contigs (see an example in S9 Fig). Finally, we also observed clades with extreme enrichment scores on trees

obtained from each of the gene regions themselves (Fig 4E; *MC1R* enriched for SA/SC and *ASIP* enriched for Ugi individuals).

We next used SIA [20], a supervised deep-learning method, to infer the strength and time of onset of selection on individual variants within the candidate regions associated with melanism on SA/SC and Ugi. Our models performed well on data simulated under the demographic parameters inferred by G-PhoCS (using msprime and SLiM), distinguishing neutral sites from those under selection, and were able to distinguish soft from hard sweeps in most cases (S10A and S10B Fig). For this task, we assigned the class with the highest probability as the predicted class, which according to the benchmark with simulated data, resulted in a false positive rate (FPR) of 6–8% when distinguishing neutral regions from those under selection (S10A Fig). We note, however, that a more stringent probability cutoff could be applied to specifically reduce the FPR for exploratory analyses such as whole-genome selection scans. When applied to the real data, SIA found evidence for soft selective sweeps on multiple variants in the peak region of contig 400, including sites associated with melanism in our GWAS (Fig 5A). We observed the strongest selection (s ~ 0.02) on the variants within and around *MC1R* (which were found on the same haplotype), with the timing of selection onset inferred to be ~500 generations before present on mutations that were ~78K generations old (Fig 5B). Although our models tended to overestimate selection coefficients when the true *s* was small, the overestimated values of *s* were typically below 0.01 (S10 Fig), which is not the case for *MC1R*. On contig 947, the sites associated with melanism in our GWAS did not vary (i.e., are fixed) in the Ugi population (S6 Fig), which may hinder our ability to detect signatures of selection when applying SIA to this population. Despite observing several signals of selection in the association peak in other analyses (e.g., high species enrichment or low RTH'), SIA inferred these sites as neutral (Fig 5C). Although among the sites identified by the GWAS we observed some towards the end of the contig which showed the highest probability of having undergone soft sweeps, they also had a $p_{neu} > 0.05$ and were therefore conservatively classified as neutral. We did however observe non-neutral sites close to *ASIP*. The site with the highest assignment probability to a given class has undergone a soft sweep (P(soft) = 0.842), is 12–14 k generations old, and was under selection (s ~ 0.01) for ~2,200 generations. Overall, *MC1R* was among the strongest and most recent targets of selection in the genome of individuals from the SA/SC population (S11 Fig).

## Discussion

Our findings show how melanism originated twice in the polyphyletic *M. c. ugiensis* from a chestnut-bellied ancestor: once on Ugi and a second time on SA/SC [9,14]. Moreover, the molecular basis of this convergent phenotype is likely to be different on each island. Our study is novel in identifying how selection has shaped the phenotype on either island, and by being able to time these events.

Black plumage on SA/SC likely originated under strong and recent selection (in the order of 1,000 years before present, assuming a generation time of 2 years) on a series of standing mutations (i.e., a soft sweep) in and around the *MC1R* gene. Selection on *MC1R* is comparable in strength to what was found for the same gene in pocket mice [21] or on the LCT gene (associated with the lactase persistence trait) in European human populations [20,22]. Our ARG based analysis dates the origin of these mutations to ~78 K generations before present (Fig 5B), which is older than the inferred split between SA/SC and Makira (~68 K generations), suggesting they could have originated in the larger Makira population and existed at low frequencies in SA/SC until the mating preference for melanic males reached a frequency threshold that triggered the recent selective sweep (e.g. [23]). Consistent with this scenario, these

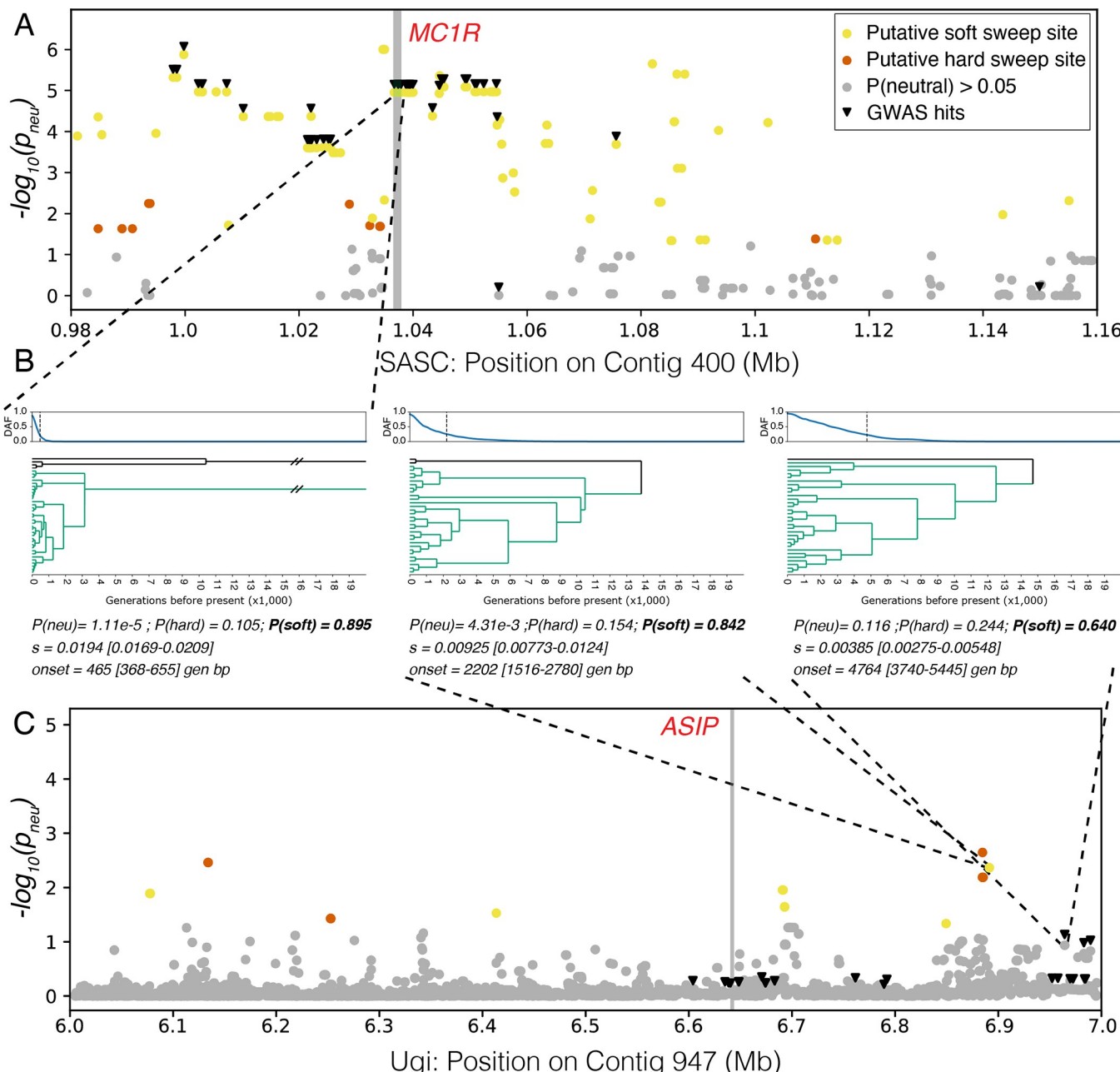

**Fig 5. Estimation of positive selection at *MC1R* in the SA/SC population and at *ASIP* in the Ugi population using SIA. A**. Negative log probability of neutrality (-$log(p_{neu})$) in the SA/SC population at candidate sites near the *MC1R* gene. GWAS hits are highlighted with inverted triangles. Significantly non-neutral sites ($p_{neu}$ < 0.05) are colored by the predicted class with the highest probability. **B**. Inferred derived allele frequency (DAF) trajectories, local genealogies and selection parameters at three loci of interest. 95% confidence intervals of inferred selection parameters are shown in square brackets. These were estimated using random dropouts at inference time (see **Materials and Methods**). The vertical dashed line on the DAF plot indicates selection onset inferred by SIA. Derived lineages are colored in aquamarine. Highlighted here are a GWAS locus in the *MC1R* coding sequence, the locus inferred to be under the strongest selection near the *ASIP* gene and a GWAS locus near the *ASIP* gene. The location of these loci is projected onto panels (**A**) and (**C**) by dashed gray lines. **C**. Negative log probability of neutrality in the Ugi population at candidate sites near the *ASIP* gene. Details of the figure are otherwise similar to panel (**A**).

derived mutations are present at low frequencies on Makira and Ugi (S4 Fig). We found among the strongest signatures of selection in the genome on the coding *Asp119Asn* mutation in this gene, a substitution that has been observed independently in other taxa and is known to constitutively activate *MC1R* [24], leading to melanism in some domestic animals [7,25,26]. Despite these lines of evidence, we can't rule out that other nearby mutations (perhaps cis-regulatory) also contribute to shaping the coloration phenotype in the SA/SC population.

Various lines of evidence suggest selection in the genomic region containing *ASIP* in the melanic Ugi population, including high species enrichment, higher levels of differentiation ($F_{ST}$), low RTH', or low nucleotide diversity. However, these statistics were calculated as windowed averages, making it hard to precisely determine the variants under selection. The coding *Ile55Thr* substitution on *ASIP* (and the other sites identified by the GWAS) was fixed on Ugi and only found in five additional individuals from the other islands (mostly in heterozygosity; S6 Fig). This association, together with the fact that mutations on the N-terminal portion of *ASIP* (where this substitution occurs) can disrupt binding and lead to melanism in other taxa [27,28], suggest a causal role. However, SIA did not infer this mutation to be under selection (or any of those identified by our GWAS analysis) and instead found other positions close to *ASIP* to be under selection. It is possible that these variants were not identified by our GWAS because of their patterns of segregation on Makira and, conversely, that SIA did not find the GWAS hits to be under selection because of their lack of variation in the Ugi population or because those events may have been too old. The estimated age of the *Ile55Thr* substitution is in the order of 174 K generations before present, and SIA was not trained to detect selection older than 20 K generations, and rarely identified selection older than 10 K generations (S10 Fig). There are a few alternative interpretations of these complex signatures of selection on contig 947. It is possible that the coding position on *ASIP* is an example of an old and completed sweep, and that SIA has detected selection on additional, perhaps cis-regulatory, and more recent mutations (dated to 12-14K generations before present with an estimated selection onset in the order of 5–10 times older compared to what was estimated for *MC1R*). These cis-regulatory mutations could modify the direct (i.e., plumage color) or potential pleiotropic (e.g., stress response, food intake) effects of the *Ile55Thr* substitution on *ASIP* [29]. Alternatively, the GWAS and SIA may have identified sites towards the end of contig 947 (between positions 6.9 and 7) that are independent of selection on *ASIP* and could contribute or be unrelated to differences in coloration. Similarly, additional mutations in other association peaks may contribute to melanism on Ugi, although we did not identify coloration genes in those genomic regions, suggesting they mediate additional phenotypes in which the Ugi and Makira populations differ. Finally, recombination in the flanking regions of hard sweeps can lead to the erroneous identification of soft sweeps (or "soft shoulders"; [30]). SIA could have identified the soft shoulder of a hard sweep on *ASIP* (Fig 5C), yet we consider this scenario to be unlikely since this erroneous classification was uncommon when using a similar approach on a different study system [19]. Overall, results from multiple population genetic approaches suggest selective sweeps occurred in the genomic region containing *ASIP* in the Ugi population.

The probability of gene reuse in parallel phenotypic evolution has been estimated to be particularly high when populations are young and closely related [31], as is the case for SA/SC and Ugi. It is therefore surprising that the older *ASIP* mutations do not also mediate melanism on SA/SC, especially since we observed one SA/SC and a few Makira individuals carrying Ugi haplotypes from the *ASIP* region (S6 Fig). One possible explanation is that gene flow between satellite islands is sufficiently low that the *MC1R* mutation swept before the *ASIP* mutations reached SA/SC.

Our findings highlight how independent selective sweeps on a receptor/ligand pair can lead to melanism on two island populations. In *M. c. ugiensis* this trait has been repeatedly favored by selection, and it remains to be determined if the same is true for other instances of island melanism [8,10,32]. There are several hypothesized benefits of darker plumage coloration, including abrasion resistance, protection from UV radiation, thermoregulation, crypsis, and parasite resistance (e.g., [33,34]). Furthermore, avian coloration is known to mediate reproductive isolation [35], especially in the early stages of speciation, and numerous incipient species have been found to differ primarily in melanin-based coloration traits [36–40]. Field experiments have shown that species recognition is mediated by plumage color in melanic and chestnut-bellied birds from Santa Ana and Makira, respectively [15,16]. Therefore, the strong selective pressures we observe in and near pigmentation genes may be the combined product of the advantages of melanic plumage and sexual selection driven by female choice. Overall, our study shows how independent mutations on individual coloration genes can lead to the convergent evolution of a phenotype that is favored on small islands, which, in turn, could promote reproductive isolation and the repeated evolution of incipient species.

## Materials and methods

### Ethics statement

All birds were caught with mist nets then measured, tagged, blood-sampled and released as part of a long-term study. Permission to collect samples and work in the Solomon Islands was granted by the Ministry of Environment, Climate Change, Disaster Management & Meteorology (BR/2014/002). Research was approved by the University of Miami Institutional Animal Care and Use Committee (IACUC) protocols number 11–116, 14–097 and 17–071.

### Sampling and dataset

A total of 57 individuals from the island of Makira and the neighboring islands of Ugi and Santa Ana/Santa Catalina (hereafter SA/SC) were included in this study from samples collected between 2006 and 2018 (S2 Table). Twenty-two Chestnut-bellied Monarch (*Monarcha castaneiventris megarhynchus*) birds were sampled from three sites on Makira: 7 birds from Waimasi directly across Ugi, 8 birds from Kirakira along the northern coast of Makira, and 7 birds from Star Harbour across from SA/SC. Thirty-five *M. c. ugiensis* birds were included from two satellite island groups: 16 birds from the island of Ugi, and 19 birds from SA/SC. Ten birds of intermediate plumage color (partial chestnut) were sampled, 6 from Makira and 4 from SA/SC. The remaining birds from SA/SC and all of those from Ugi were melanic, while those from Makira were all chestnut-bellied except for one melanic individual. Finally, a sample from a chestnut-bellied *M. c. obscurior* bird caught in the Russell Islands in 2013 was used to sequence and assemble the reference genome, and five additional individuals caught in 2012 were resequenced as outgroups for phylogenetic analysis.

### Reference genome assembly and annotation

We assembled and annotated a genome from a male Chestnut-bellied Monarch (*Monarcha castaneiventris obscurior*) sampled in the Russell Islands (Solomon Islands; individual RU430). We obtained both short-read Illumina data and long-read Pacific Biosciences (PacBio) data from the same individual, and all sequencing was conducted by Novogene Co. A fragment library was prepared using the NEBNext DNA Library Prep Kit with an insert size of 350 bp, and paired-end sequenced on an Illumina Novaseq 6000 machine, producing 168 gb of raw data (approximately 140x coverage). PacBio SMRTbell libraries were prepared and sequenced

on 5 flow cells of the Sequel platform, generating 71.4 gb of data (approximately 60x coverage) with 7 million subreads with an average N50 of 15.5 kb. We used Samtools version 1.11 [41] to merge the subreads from the five flow cells, filter out subreads that were shorter than 4.5 kb (retaining 67.6% of the subreads), and to convert the file to fasta format. We performed the genome assembly with MaSuRCA version 3.3.3 [42], an assembler which can incorporate both Illumina and PacBio data by being able to use reads of variable lengths. We produced assembly statistics with Quast version 5.0.2 [43], obtaining a total assembly length of 1.08 gb distributed in 899 contigs, with an N50 of 20.2 mb and 2.1 Ns per mb. We assessed the completeness of our reference assembly by searching for the Passeriformes set of 10,844 single copy orthologs using BUSCO version 5.1.2 [44]. Our reference genome contained a complete copy of 95.7% of the orthologs in this gene set, 95.4% were found as single copy genes and 0.3% were duplicated. There was a total of 3.4% of these genes that were missing from our assembly and an additional 0.9% were found fragmented. We estimated the chromosomal location of the 899 contigs in our assembly by aligning them to the chromosome level Zebra Finch genome (bTaeGut2.pat.W.v2 downloaded from http://www.ncbi.nlm.nih.gov/) with the *Chromosemble* function from the Satsuma version 3.1 pipeline [45], and assigning contigs to the chromosome with the top hit. This function also provides a version of the reference genome with contigs aligned and oriented into pseudochromosomes, assuming synteny between the Chestnut-bellied Monarch and the Zebra Finch. We conducted downstream analyses with both versions of the reference genome and obtained equivalent results (e.g., the same association peaks in our GWAS), so we decided to present those based on the version of the genome that does not assume synteny with the distantly related Zebra Finch.

We annotated the reference genome by first generating a library of the repetitive sequences with RepeatModeler version 2.01 [46]. These simple and complex (e.g., transposible elements) repeats can be subsequently masked to avoid being incorreclty annotated as genes from the organism of interest. We then ran two iterations of the MAKER pipeline version 3.01 [47] to produce gene models. The first iteration generated gene models by training algorithms with data from Zebra Finch transcript and protein databases (downloaded from the bTaeGut2.pat. W.v2 assembly). The models are subsequently refined during a second iteration of the pipeline that uses the output of the first MAKER run as input. In total the pipeline produced 15,226 gene models (72.3% of the 21,049 genes annotated for the Zebra Finch).

## Population level genome sequencing and variant discovery

We sequenced the genomes of 57 individuals, 22 belonging to *M. c. megarhynchus* sampled on the island of Makira and 35 to individuals of *M. c. ugiensis*, 16 of which were sampled on the island of Ugi and 19 sampled on the islands of SA/SC. We extracted DNA from blood samples using the DNEasy blood and tissue kit (Qiagen, CA, USA) and libraries were prepared by Novogene Co with the NEBNext DNA Library Prep Kit, with an inset size of 350 bp. Sequencing was performed on an Illumina Novaseq 6000 machine by Novogene Co, obtaining 5,967 million paired end, 150 bp reads. Based on the number of raw (pre-filtering) reads, we expected the depth of coverage to range across all individuals from between 21.5 and 36.8x (average of 26.2x).

We first assessed the quality of individual libraries using fastqc version 0.11.8 (http://www. bioinformatics.babraham.ac.uk/projects/fastqc) and performed quality filtering and trimming, adapter removal and merged overlapping paired end reads with AdapterRemoval version 2.1.1 [48]. Once reads were filtered we proceeded to align them to the reference genome using Bowtie2 version 2.4.3 [49] using the very sensitive local option, which resulted in an average alignment rate of 99.4%. We manipulated the alignment files using Samtools version 1.11 [41], converting *sam* files into *bam* format and sorted and indexed them. We used Picard Tools

version 2.8.2 (http://broadinstitute.github.io/picard/) to mark PCR duplicates, GATK version 3.8.1 [50] to realign around indels, and finally Picard Tools to fix mate-pairs. We obtained an average depth of coverage of 26.3 +/- 4.5x and an average duplication rate of 21.4 +/- 1.6 by computing alignment statistics using qualimap version 2.2.1 [51].

Our genotyping pipeline started by producing individual genomic variant call files for each sample with the "Haplotypecaller" module from GATK, and we subsequently used the "GenotypeGVCFs" module to summarize variants into a single variant file for the entire dataset. We selected SNPs with the "SelectVariants" module of GATK and retained those that satisfied the following filters: $QD < 2$, $FS > 60.0$, $MQ < 30.0$, $ReadPosRankSum < -8.0$. Finally, we used VCFtools version 0.1.16 [52] to retain 4,799,460 variant sites present in at least 80% of individuals, with mean depth of coverage between 2 and 50 and a minor allele count of at least 8 (equivalent to a minimum of four homozygote individuals, which represents 25% of the population with the smallest sample size). We used this dataset for downstream analyses unless otherwise stated.

## Population structure, genetic differentiation and summary statistics

We assessed population structure and admixture among individuals in our sample by conducting a Principal Component Analysis (PCA), constructing an admixture plot and building a Maximum Likelihood tree. We also quantified differentiation by calculating $F_{ST}$ values among the populations from the three sampled islands. The PCA was conducted in R version 4.0.2 [53] with the package SNPRelate version 3.3 [54]. We assessed structure and admixture using the program Admixture version 1.3.0 [55]. For this analysis we first thinned the dataset to avoid including linked SNPs with VCFtools, retaining 101,076 SNPs that were at least 10 kb apart. We manipulated the vcf file in VCFtools and plink version 1.9 [56] to convert it to bed format and ran Admixture with a K of three populations. We also ran Admixture analyses exclusively for the two focal contigs with association peaks, in 100 kb sliding windows. We ran the analysis separately for Makira vs SA/SC individuals on contig 400 and Makira vs. Ugi on contig 947 (i.e., K = 2). We plotted these values by using a smoothing line in ggplot2 [57]. To build a tree we first re ran the pipeline described in the previous section using identical parameters, but including five outgroup *M. c. obscurior* individuals sampled in the Russel Islands (>330 km away). This iteration of the pipeline produced 5,811,866 SNPs, 5,094,873 of which (those that had the minor allele in homozygosity in at least one individual) could be used to build a tree using RAxML version 8.2.4 [58]. We implemented the "ASC_GTRGAMMA" model in combination with the Lewis correction for ascertainment bias, and carried out 200 bootstrap replicates. We used VCFtools to calculate $F_{ST}$ values for non-overlapping 5 kb and 25 kb windows, and subsequently obtained average values and standard deviations for each contig/population comparison in R. We also calculated Tajima's D and nucleotide diversity (π) in non-overlapping, 5 kb windows, with VCFtools (independently for each population) using a dataset without the minor allele frequency filter (see the section on Demographic reconstruction). Additionally, we calculated the haplotype-based statistics H1, H2, H12 and H2/H1, which are designed to distinguish between soft and hard sweeps, using the package SelectionHapStats [59]. We obtained these statistics for non-overlapping windows of 100 SNPs, merging haplotypes with only one difference (—distanceThreshold 1) and using the dataset without the minor allele frequency filter. Finally, we used the information of the chromosomal location of each contig (based on the results from *Chromosemble*, see above) to plot $F_{ST}$ estimates obtained from autosomes and the Z chromosome separately, as values from the latter chromosome tended to be higher. We only plotted values for contigs that were at least 150 kb (six non-overlapping windows).

We also built minimum spanning networks in PopART 1.7 [60,61] from mitochondrial genomes. We first assembled mtDNA genomes from our filtered reads with MITObim 1.9.1 [62], using the "quick" option and up to 40 iterations with the full mitochondrial genome from the Hooded Crow as a template (*Corvus cornix cornix*, GenBank number NC_024698.1). We subsequently aligned the 57 individual sequences in Geneious version 10.2.6 [63] and imported the alignments into PopART 1.7. We repeated this process restricting the analysis to the COI gene alone, which is commonly used for species identification [64].

## Demographic reconstruction

We conducted demographic reconstructions using G-PhoCS version 1.3 [65] which implements an isolation-with-migration model, obtaining estimates of effective population sizes, splitting times and bi-directional migration rates. Because of the computationally intensive nature of this analysis, we conducted two separate demographic reconstructions, one including individuals from Makira and Ugi and the second with individuals from Makira and SA/SC. We also subsampled our dataset, retaining 7 individuals per island (we did not include individuals with intermediate coloration or the melanic individual from Makira). We re-exported 11,537,213 SNPs without a minor allele frequency filter to avoid biasing our analysis by only using data including alleles segregating at higher frequencies, and used these SNPs to generate sequence files for each individual with the "FastaAlternateReferenceMaker" module in GATK. We subsequently sampled for each individual 1,700, 1 kb sequences at intervals of at least 100 kb from autosomal contigs that were larger than 1 Mb. We ran the multi-threaded version of the program for 2 million iterations, discarding the initial 100,000 as burn-in, and estimated 6 demographic parameters in each of our two models (three effective population sizes, one splitting time, and two migration rates). We checked that the traces from the different parameter estimates were stationary and that the effective sample sizes were large (range: 228–9020) using the coda package in R [66]. To convert median and 95% Bayesian credible intervals for each parameter from mutation scale to generations or individuals we used an approximate mutation rate estimate of $10^{-9}$ per bp per generation [67]. We note that the assumption of mutation rate will impact the absolute estimates of population sizes and divergence times produced by the model, however we try to focus our interpretations on relative comparisons which are independent of the assumed mutation rate. The number of migrants per generation, which is independent of the assumption of mutation rate, was calculated as the mutation scaled per generation migration rate times a fourth of the theta parameter for the receiving population ($m_{a>b} \times theta_b/4$).

## Genome wide association analysis (GWAS) and identification of genes in divergent regions

We conducted a phenotype-genotype association analysis using the Wald test implemented in Gemma version 0.98.4 [68]. We generated a phenotypic variable in which chestnut individuals were scored as 1, fully melanic individuals were scored as 2, and intermediate individuals as 1.5. The GWAS tests the association between this phenotypic variable and SNP genotypes by fitting univariate linear mixed models, which account for population structure by calculating and including an inter-individual relatedness matrix among all samples as a covariate. We conducted two analyses, one including individuals from Makira and SA/SC and a second with individuals from Makira and Ugi, as we had previous evidence indicating that each island had a different origin of melanism [9]. We did not conduct a GWAS comparing SA/SC and Ugi individuals as these two populations are not sister and have pronounced population structure. We corrected for multiple tests by using the total number of comparisons conducted across

both GWAS (Makira vs. SA/SC and Makira vs. Ugi), and used this conservative α threshold to assess significance (α = 0.05/(2*4.7) M SNPs ~ 5.3x10-9). We subsequently visualized our results by log-transforming the p-values, changing their sign, and building Manhattan plots with the R package qqman [69]. SNPs showing statistically significant associations tended to cluster together in groups (generally more than 5 SNPs) which we defined as association peaks. In other cases, we also observed single or at most a couple of isolated SNPs beyond the level of statistical association which we did not treat as association peaks. We searched for the genes contained in the association peaks by inspecting these regions in the annotation file using Geneious version 10.2.6 [63] and compiled a list of gene models within each region. We subsequently obtained information on these annotations of interest from the NCBI database (http://www.ncbi.nlm.nih.gov/). We explored the relationship between genotypes at different loci within each association peak by phasing and imputing missing data using BEAGLE version 3.3.2 [70]. This resulted in two haplotypes per individual for each peak with which we calculated a distance matrix in the R package vegan [71] and plotted it with the function phylo. heatmap() from the R package phytools [72]. We also calculated linkage disequilibrium (LD) between different sites by computing R2 values in VCFtools.

## Ancestral Recombination Graph (ARG) inference and derivation of ARG-based statistics

We generated statistics derived from ARGs as described in detail in [19] using scripts deposited in GitHub (https://github.com/CshlSiepelLab/bird_capuchino_analysis). We first inferred ARGs for two contigs with association peaks (contig 400 and contig 947; total of ~9 Mb) and 20 similarly-sized contigs (ranging from 1.2 to 11 Mb; total of ~65 Mb) that did not contain association peaks. We inferred ARGs using the arg-sample module from ARGweaver version 1 [18], which estimates a local tree for each position along the contig. We ran the software independently on each of the 22 contigs indicating that the data were unphased and assuming a mutation and a recombination rate of $10^{-9}$/bp/gen [19,67]. We set the effective population size to 500,000 individuals and the following options for the remaining parameters required by the software: -c 5—ntimes 20—maxtime 1e7—delta 0.005—resample-window-iters 1—resample-window 10000 -n 1000. We sampled the last of 1,000 MCMC iterations and used it to extract a local tree at intervals of 500 bp, discarding the edges of each ARG block (initial and final 50 kb) where there is uncertainty in the inferred topologies. We subsequently calculated two statistics from each tree for downstream analyses: species enrichment scores and RTH' (see [18] for RTH and [19] for a modification in how we normalize TMRCA to obtain RTH' or S7A Fig).

Species enrichment scores are defined as the probability of observing a subtree with n leaves for which k are mapped to a certain species or population, assuming a hypergeometric distribution. Therefore, if a local tree contains a large clade composed of individuals from the same species this will be reflected in a high enrichment score for that species. Because any given tree contains various subtrees, the score for each species at each site was defined as the highest score obtained from all the possible subtrees. RTH' was calculated by dividing the time to the most recent common ancestor of half the haploid samples for a given species (TMRCAH; $n_{Makira}$ = 22, $n_{SA/SC}$ = 19, $n_{Ugi}$ = 16) by the age of the youngest subtree that contained at least half of all haploid samples (n = 57). The benefits of this normalization are that it is sensitive to various types of selective sweeps (e.g., partial sweeps, those shared by multiple species or complete species-specific sweeps) and that it is independent of the variation in coalescent times that is observed along the genome [19]. We obtained 20 kb nonoverlapping window values for each statistic, derived from averaging statistics obtained from 40 individual trees. For each species,

this process produced windowed averages for species enrichment and RTH', for the two contigs with association peaks and the sample of 20 contigs that did not contain association peaks.

We assessed statistical significance by generating empirical distributions for each parameter from the total of 3,682 20 kb windows. We established species and parameter specific significance thresholds by finding the cutoff value that represented the top (species enrichment) or bottom (RTH') 0.01, 0.005 and 0.001 of the distribution. Cutoff values that defined different slices of the distribution were used to establish statistical significance at different P values. Windows that fell beyond or below a threshold (e.g., P<0.001) were considered to come from a region of the genome with clades that are statistically significantly enriched in a given species, or to have statistically significantly shallow clades (RTH'), respectively. Finally, we exported randomly selected individual topologies or trees which illustrated extreme enrichment or RTH' values for particular species, for the regions containing association peaks or for specific genes.

## Inference of positive selection

The analyses of species differentiation and cross-species ARG statistics are useful tools to detect signals of positive selection in genomic windows. To further localize the target of selection and infer parameters of positive selection such as the selection coefficient, time of selection onset and allele frequency trajectories, we employed the machine learning method implemented in SIA ("Selection inference using the ancestral recombination graph") [20]. SIA uses a Recurrent Neural Network to leverage features of single-population genealogies. Selection in a population leaves characteristic signals in its genealogy that can be picked up by SIA to make inferences of selection parameters for individual variants that map to gene trees embedded in an ARG (S7B Fig).

We simulated data for training and benchmarking the SIA model by initializing neutral simulation in msprime [73] and continuing simulation of positive selection in SLiM [74,75], to maximize computational efficiency. We ran coalescent simulation in msprime up to the generation of selection onset (or in the case of neutral simulations, a randomly sampled generation), saved the progress in tree sequence format, and loaded the tree sequence in SLiM to carry on with forward simulation. We simulated separate datasets for the SA/SC/Makira and Ugi/Makira population pairs, each under a two-population, 5-parameter demographic model inferred by G-PhoCS (see above and Fig 1G), with effective population size scaled down by 10-fold. Scaling down the population sizes reduces the running time of the simulations but requires scaling other parameters accordingly. Because the migration rates from each of the satellite islands into Makira were inferred by G-PhoCS to be negligible, these were ignored in the simulations. However, we simulated gene flow from Makira into each of the satellite island populations because it was inferred to be much higher (Fig 1G) and can have a non-trivial effect on selection inference. For example, a completed hard sweep followed by subsequent introgression and recombination of the ancestral haplotype could be mis-classified as soft by a model trained without simulations of such a scenario. For sweep simulations, selection coefficients (*s*) were sampled between 0.001 and 0.025 (scaled up to 0.01–0.25 for simulations) from an equal mixture of a uniform distribution and a log-uniform distribution. We kept only sweep simulations where the current derived allele frequencies at the sweep site was greater than 0.2 and allowed for alleles that are "recently fixed". This sampling scheme corresponded roughly to a range of selection onset from 250 to 20,000 generations before present, a regime in which SIA would be trained to detect positive selection. For soft sweep simulations, the allele frequency threshold ($f_{init}$) above which selection acts on the allele was sampled uniformly by $f_{init} \sim U(0.01, 0.1)$. To simulate a soft sweep, at the generation of selection onset, we picked

a random clade of the satellite island population whose size matches the sampled $f_{init}$. We then added a mutation to the branch leading to the MRCA of this clade before turning on selection at this variant. This particular MRCA could be a native (such that the mutation occurred on the satellite island), or alternatively a migrant from Makira (such that the allele came from standing variation in the Makira population). Nevertheless, since SIA uses features of single-population genealogies of the satellite island population, it is agnostic to the two scenarios. Each dataset consists of 1,500,000 neutral, soft and hard sweep simulations of 100kb regions (equal split among the three categories). For sweep simulations, the sweep site was at the center of the region. The datasets were used to train and benchmark two separate SIA models for the SA/SC and Ugi populations following a train-val-test split of 85%-5%-10%. The ARG inference process, genealogical feature extraction and deep learning architecture used for building the SIA model are described in detail in [20]. A cartoon illustration of the genealogical features is provided in S7B Fig. We applied the SIA model to detect signals of positive selection in the SA/SC and Ugi populations using the dataset without the minor allele frequency filter, and for putative sweep sites, we inferred selection coefficients and the time of selection onset. To gauge the uncertainty of the parameter estimates, we applied dropout to the trained SIA model at inference time [76]. We ran the model 1,000 times, each with random sets of dropout nodes, to obtain 1,000 samples of the model prediction from which a 95% confidence interval was derived. We conducted these predictions across the 190 scaffolds that were longer than 100kb and had at least 1,000 called variants (without applying a minor allele frequency filter). Sites with a probability of being neutral greater than 0.05 were considered to be neutral. In addition, for particular sites of interest, we applied the model to infer allele frequency trajectories. Finally, we dated the origin of several mutations by estimating the midpoint age of the branch in which they first appeared.

## Supporting information

**S1 Table. Genes within the association peaks.**
(DOCX)

**S2 Table. Details for the samples used in this study.**
(DOCX)

**S1 Fig.** Mitochondrial minimum spanning networks. **A**. Haplotype network based on a ~17 kbp alignment of the mitochondrial genome. **B**. Haplotype network based on 650 bp of the mitochondrial COI gene, commonly used for species identification. Branch lengths are proportional to the number of nucleotide differences between haplotypes, which are indicated by short lines on each branch (omitted for simplicity in the case of the full mitochondrial network).
(TIF)

**S2 Fig.** Linkage disequilibrium among association peaks identified in the GWAS conducted with Makira and Ugi individuals. Average (below the diagonal) and maximum (above the diagonal) R2 values among all the statistical outlier sites in the GWAS from different pairs of association peaks. The chromosome and contig to which each peak belongs is indicated in red, and the size and color of the circles denotes the magnitude of LD.
(TIF)

**S3 Fig.** Genetic differentiation within and outside of association peaks. **A and B**. $F_{ST}$ values calculated for 5 kb windows inside and outside association peaks. The plot for contig 400 compares individuals from SA/SC and Makira, while the plot for contig 947 compares individuals

from Ugi and Makira. The red dot denotes the window containing the *MC1R* and *ASIP* genes. **C to F**. Smoothed ancestry values across contigs, with association peaks indicated between vertical lines. Ancestry values were calculated in 100 kb sliding windows in Admixture, and the analysis was restricted to Makira vs. SA/SC for contig 400 and Makira vs. Ugi for contig 947. The plot in (**C**) shows increased separation between individuals from Makira and SA/SC in the peak region on contig 400 and extending approximately 0.5 Mbp in each direction. The plot includes only individuals from Makira and SA/SC that were homozygotes for the *Asp119* or *Asn119 MC1R* mutation, respectively. The plot in (**D**) shows the six individuals from Makira and the four individuals from SA/SC which were heterozygotes for this mutation and had intermediate plumage, and shows overall more admixture than what is seen in (**C**). **E**. Ancestry values across contig 947 showing increased resolution in the peak region and extending approximately 2 Mb downstream. Three individuals from Makira which were heterozygotes for the *Tre*55 *ASIP* mutation are labeled in gray (only one of these individuals had intermediate plumage). The single melanic individual from Makira (MA705), which was homozygous for the derived *Tre*55 mutation, is labelled in black. For all four individuals, Ugi ancestry decreases to levels comparable to other Makira individuals about 1 Mb downstream of the peak. **F**. Ancestry across contig 27 shows little resolution compared to the association peaks on contigs 400 and 947. We note that ancestry values range from 0 to 1, but that the plots extend beyond this range because of the smoothing algorithm and particularly the uncertainty shown by the confidence bands. Values beyond the [0,1] interval are therefore meaningless. (TIF)

**S4 Fig.** Clustering of haplotypes obtained from the association peak on contig 400. Phased genotypes for the 61 SNPs located in the association peak on contig 400. Rows represent single chromosomes, therefore individuals are represented twice in the clustering tree on the left. The four nucleotides, the collection locality and the coloration phenotype are color-coded as indicated at left. The three SNPs within the *MC1R* coding region are indicated with a black rectangle. All individuals from SA/SC contained at least one haplotype in the region delimited by the SNPs with significant association scores around *MC1R* that differed from the one present in most individuals from Makira and Ugi. All the melanic individuals from SA/SC possessed two copies of this haplotype, while the four individuals with intermediate coloration possessed one of each, as was the case for two of the six individuals with intermediate coloration from Makira. All melanic individuals from SA/SC carried two copies of the derived *Asn119* mutation, while all but one of the chestnut-bellied individuals from Makira had two copies of *Asp119*. The individuals with intermediate coloration (from either Makira or SA/SC) were heterozygotes for this coding mutation. The exception to this pattern was a single chestnut-bellied individual from Makira (MA714), which was a heterozygote yet was scored in the field under heavy molt and may have been incorrectly classified as having a chestnut belly. Finally, the derived *Asn199* mutation existed primarily on the haplotype background found on SA/SC, but also to a lesser extent on the haplotype background found on Makira and Ugi. We note that the chestnut-bellied MA714 bird carried the *Asn119* mutation on the most common haplotype background observed on Makira. The birds from Ugi, which are all melanic, were sometimes homozygous for either allele or heterozygotes. (TIF)

**S5 Fig.** Principal component analyses derived from the SNPs within the association peaks. PCAs from the variants within the association peaks on contig 400 and contig 947. Coloration phenotype and the island where individuals were sampled are color-coded. (TIF)

**S6 Fig.** Clustering of haplotypes obtained from the association peak on contig 947. Details as in S4 Fig. Ugi individuals had two copies of a haplotype that was different from the one present in SA/SC and Makira individuals. The only melanic individual from Makira (MA705) also carried two copies of the derived Ugi haplotype. We found a few heterozygote individuals but there was no obvious pattern with respect to their coloration.
(TIF)

**S7 Fig.** Statistics and feature encoding of the genealogy. **A.** RTH' tests for the reduction in within-species TMRCA and is defined as the ratio between the TMRCA of half of the samples from *a given species* and the age of the youngest subtree that contains at least half of *all* samples. The species enrichment score tests for species differentiation in local trees and is defined as the maximum score associated with a given species in a subtree of the full genealogy. For a subtree, the species enrichment score is the probability of observing the number of samples of a particular species in that subtree under a hypergeometric distribution. Here the coloring of the leaves indicates hypothetical species and the example illustrates the statistics with respect to the green species. **B.** Genealogical features for the SIA model consist of the number of lineages in the genealogy at a set of discrete time points ($t_0$, $t_1$, . . .). The time points are chosen in an approximately log-uniform manner resulting in finer discretization of more recent time scales. In addition, when encoding the genealogical features at a particular site of interest, we encode separately the counts of ancestral (shown in black in the example) and derived (shown in aquamarine) lineages. The aquamarine cross indicates the branch where the mutation occurred.
(TIF)

**S8 Fig.** Highest species enrichment scores and lowest RTH' values for peak regions on contigs 400 and 947. Representative trees from the peak regions which show extreme values of species-enrichment (**A**) and RTH' (**B**) for SA/SC and Ugi.
(TIF)

**S9 Fig.** Species enrichment scores and RTH' values for a control contig. Plots showing species enrichment (**A**) and RTH' (**B**) in 20 kb windows along contig 27. Horizontal lines show species-specific levels of statistical significance ($p < 0.005$). Trees obtained from random positions on contig 27 (each position is shown under the tree) indicating species enrichment scores (**C**) and RTH' (**D**) values for each population.
(TIF)

**S10 Fig.** Benchmarking of the SIA models. **A**. Confusion matrices generated by applying the SA/SC and Ugi models to simulated data. Instead of applying a specific probability threshold, the predicted class was identified as the one with the highest probability according to the model. Under this maximum likelihood classification scheme, both models perform best distinguishing neutral from selected sites, and moderately when identifying soft or hard sweeps. **B**. One-versus-rest (OvR) receiver operating characteristic (ROC) curves of the model classification performance on simulated data. The models perform very well on distinguishing sweeps, and less so on precisely identifying hard or soft sweeps. **C**. Comparisons between true and inferred selection coefficients and time of selection onset (expressed in generations before present) for SA/SC and Ugi models trained to detect soft or hard sweeps. Models trained to predict hard sweeps generally perform better than those trained to predict soft sweeps, which tend to overestimate small selection coefficients and recent selection onset times.
(TIF)

**S11 Fig.** Genome-wide estimates of selection coefficients and time of selection onset. Predictions are based on all the variants from the 190 scaffolds longer than 100kb.
(TIF)

**S1 Data.** File containing the numerical data underlying the figures in the manuscript.
(XLSX)

## Acknowledgments

We thank the Fuller Evolutionary Biology Lab and Cornell's BioHPC Cloud for access to computing resources. We thank Dr. Elsie Shogren, Dr. Rebecca Safran and Dr. Daven Presgraves for comments on a previous version of this manuscript.

## Author Contributions

**Conceptualization:** Leonardo Campagna, J. Albert C. Uy.

**Data curation:** Leonardo Campagna, J. Albert C. Uy.

**Formal analysis:** Leonardo Campagna, Ziyi Mo, J. Albert C. Uy.

**Funding acquisition:** J. Albert C. Uy.

**Investigation:** Leonardo Campagna, Ziyi Mo, Adam Siepel, J. Albert C. Uy.

**Methodology:** Leonardo Campagna, Ziyi Mo, Adam Siepel, J. Albert C. Uy.

**Project administration:** Leonardo Campagna, J. Albert C. Uy.

**Software:** Ziyi Mo, Adam Siepel.

**Visualization:** Leonardo Campagna, Ziyi Mo.

**Writing – original draft:** Leonardo Campagna, J. Albert C. Uy.

**Writing – review & editing:** Leonardo Campagna, Ziyi Mo, Adam Siepel, J. Albert C. Uy.

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
