## [Decision Letter · Decision Letter 0]

11 Aug 2022

Dear Dr Campagna,

Thank you very much for submitting your Research Article entitled 'Independent selective sweeps on pigmentation genes mediate parallel evolution of island melanism in two incipient bird species' to PLOS Genetics.

The manuscript was fully evaluated at the editorial level and by independent peer reviewers. The reviewers appreciated the attention to an important topic but identified some concerns that we ask you address in a revised manuscript

We therefore ask you to modify the manuscript according to the review recommendations. Your revisions should address the specific points made by each reviewer.

[LINK]

Yours sincerely,

Nicolas Bierne

Academic Editor

PLOS Genetics

Kirsten Bomblies

Section Editor

PLOS Genetics

Dear Dr. Campagna and Dr Uy,

Thank you for your submission to PloS Genetics. We have received three thoughtful reviews of your manuscript entitled “Independent selective sweeps on pigmentation genes mediate parallel evolution of island melanism in two incipient bird species”. The referees are globally positive and expressed mainly minor concerns, though important to account for. I’d ask you to account for all these concerns in a revised version. Two referees were surprised you did not investigate further, nor discuss, the other peaks showing up in the GWAS, and especially the one of the Z chromosome. Referee 2 asks for more details about SNP filtering. Referee 3 suggests additional analyses that you can do, or alternatively you can argue why you did not. They all three provide additional comments that should help you to clarify some points of the paper. The paper has been written to be short but now it has been transferred to PLOS Genetics you have more space to explain complex analyses in more detail.

On my side, I also was very interested by your work and I learned a lot with the reading of your ms and the literature I had to catch up to understand it. It is well written and the results are important findings. The method is appealing. I would ask you the followings:

- Please clarify the novelty of your finding. Given it was already strongly suspected that two different genes were involved in melanic evolution in the different satellite islands, independence is rather a confirmation than a new result. In addition, the evidence for a selective sweep at ASIP on Ugi is rather weak (and/or not clearly discussed). The novelty is on the evidence of selection, but mainly at MCR1 on SA/SC, and also in the method. I think the title is somewhat misleading (the “s” in “sweeps”) and could be reworked to better fit the originality of your work and still attracts readers.

- Please clarify what you mean with “independent” and “parallel” and use these terms with care. Given the molecular mechanism is different in Ugi and SA/SC, independence is obvious and only phenotypic evolution is parallel not molecular evolution. For instance the title on line 165 “MC1R and ASIP show signatures of parallel selective sweeps”, is an unusual use of the term “parallel” (although possible if well defined/explained, given “parallel” is often a catch-all term).

- The novelty is in the method and the data type used (ARG-derived) while you devoted few space to explain them, relying on previous publications. I think a cartoon (e.g. as a sup file) is needed to explain “species enrichment” and “RTH’”, in addition to the example of figure S6. Species enrichment could be called differently as you use it for population genetics, and it could be described with other words (eg “population” instead of “species”). I didn’t immediately understand what you meant with species here. This ARG-derived statistics is calculated for a sample/population, but depend on the all ARG on the whole sampling as far as I understood. It seems similar to a population-specific Fst, and somehow informs about genetic differentiation (hence the peak at ASIP in fig 3). “RTH’” (please recall RTH means “relative TMRCA half-life”) is well suited to identify diversity trough but I find it difficult to have intuition about it’s behavior as, again, the coalescence of all haploid samples is included in the calculation. SIA is a new method you developed and has not been often used. It would need more explanation. Especially, what exactly are these features from the local genealogies extracted from the ARG? Given it’s not super clear in the SIA paper, I would find it useful for the reader if you could better describe them (again a cartoon as sup file?). In addition, you said “SIA uses features of single-population genealogies of the satellite island population” (line 553) and I’d like you to better explain and elaborate on the possible implications (e.g. in comparison with the study of “species enrichment” and “RTH’” in the previous analysis).

- I was pleasantly surprised to see you used two-pop simulations in SIA. The reason is that local hard sweep in a subdivided population could easily be classified as soft sweep if a classifier is trained with single pop simulations (after fixation unswept haplotypes introgressed from the continent population into the shoulders of the swept region of the island population). I think you might insist more on this point.

- I found the discussion of the ASIP sweep (if a sweep really happened) on Ugi a little confusing. SIA prefers a neutral model at ASIP, so you discuss the little evidence of soft sweeps 300-400 Kb away in the flanking regions. What’s the idea? Regulatory regions as proposed by referee 3? what’s the differentiation? not as strong as at ASIP according to species enrichment (indeed a good old Fst Manhattan plot could be useful here, as sup file). Or is this the shoulder of the ASIP sweep (hence a soft sweep signal)? How does this region recombine? Could a chromosomal rearrangement be suspected (aslo suggested by referee 3)? If you discuss the age of the soft sweep it means you have selection on these flanking regions, not at ASIP. Please clarify each hypothesis, explaining when you consider selection is targeting Ile55Thr and when you consider selection is targeting other variants away on the contig. In addition, is an old fixation at ASIP-Ile55Thr and footprint lost compatible with low RTH’ at ASIP? Looking at the zoom around Ile55Thr in figure S5 one would rather bet on a recent hard sweep.

- Figure S7: “Circles represent statistics derived from 100-SNP windows (left panels) or 5 kb windows (right panels)” Are left panels lacking?

- Discussion: Melanic mutants segregate with low or negative s during ~100K generations and then s switch to 2% 500 generations ago. Why melanic evolution takes place now? What happens with these satellite islands, are they new? Or why would sexual selection switch on suddenly? Does melanic evolution take place recurrently in many systems and you arrived at the good pace now in this system? I'm lacking of context here.

Overall, these are matter of clarification and discussion and I'm convinced you will address my and referees concerns easily in a new version. I’m looking forward to reading it.

Best regards,

Nicolas Bierne

Reviewer's Responses to Questions

**Comments to the Authors:**

Reviewer #1: Leonardo Campagna and colleagues build a new reference genome and collect resequencing data of different Chestnut-bellied Flycatcher populations in the Solomon Islands to study the origin and evolution of melanic phenotypes. They mostly build from a previous study (Uy et al. 2016. Proc B) that already demonstrated the unique evolution of melanism in the two separate island populations. However, in this study, they identify the selective sweeps around the causative loci and from these sweeps estimate strength and timing of selection.

I really enjoyed reading this manuscript. It is clearly written and it presents interesting results. For example, the authors use nifty statistics to identify patterns of selection and use machine learning (trained with their demographic inference models) to classify soft and hard sweeps as well as the timing and strength of selection. To me, that is novel and this study will provide guidance for other studies to do this. As the authors note themselves, it was also surprising to me that at the geographic scale, the potential dispersal distances of the birds, and with the mutations being present at low frequency in Makira that the satellite populations adapted using different mutations. I think this study is thus a compelling example that shows that evolution from standing genetic variation does not have to be the default.

My main comment is that I am still curious about the other peaks showing up in the GWAS, including the Z chromosome. Is the lack of any color pattern genes in the additional GWAS peaks enough argument to dismiss them? What if these included genes that have not been linked to color or well characterized? If so, it might be important to discuss the additional peaks further.

L120 and L148 describe outlier results on the Z chromosome. This chromosome may have a proportionally important role in speciation, but no further mention is made or explanation is given.

Additional comments:

The results are mostly clear, but some parts, I think, would benefit from adding some details on the used methodology (see below for specific sections), rather than having to look for them in the methods.

L54-55: These are examples of insular systems. This sentence might benefit from adding a motivation of focusing on insular systems.

L66: Please clarify the data used in the phylogenetic analysis.

L125: Please provide some details on the demographic reconstruction here. What tool and what type of data was used?

L130: How was gene flow inferred? Please elaborate on what type of signals are looked for in the data to provide confidence in these estimates.

L183-184: “resulting from” instead of “derived from”? Or “statistical outliers” instead of “outlier statistics”?

L193: How was this data simulated, in short? (I would at least mention the program and that parameters were derived from the demographic inference model).

L198-200: Can confidence intervals be provided for the selection strength and time estimates?

L421: Please provide some more information on how G-PhoCS works.

L469: Do the two haplotypes explain the soft sweep?

L537: What is the motivation of down sampling the effective population size? I assume computational feasibility? Could this not strongly impact the training data and inferred results on selection strength and timing?

Reviewer #2: In this manuscript the authors identify and characterize independent soft selective sweeps associated with a melanism in Monarcha castaneiventris ugiensis. They show that the origin and timing of the onset of selection for a blabk belly differ between the subpopulations in the islands of Ugi and Santa Catalina / Santa Anna. This is an original study and it is noteworthy that this is a case of convergent selection although the two melanic populations share a common origin. The conclusions of the study rely on strong genetic evidence and I have a single minor comment about the methodology (my point 1)

1- SNP filtering may affect the results and the statistics: the authors sometime discarded SNPs with an absolute MAF (minor allele frequency) lower than 8. In some cases this filtering was not applied but this choice is not clearly stated. This filtering affects nucleotide diversity (pi) and the expected distribution of Tajima's D (which will be skewed towards positive values) this may partly explain the results shown on fig. S7.

2- Sampling: The sample described lines 283-292 is not clear. Please double check that the text conforms to the suppl. table s1.

3- Genetic differentiation of M. c. megarhyncus and M. c. ugiensis: Most of the results stand in the PCA analysis (fig 1D) where the first axis clearly separates the two M. c. ugiensis populations. It is also clear that Ugi is more distant than SA/SC from Makira, which is confirmed by the FST values (Fig 1F). Note that FST are computed between populations not between individuals (lines 381-382 in the main text).

4- Figures: Notice that the figures are often tiny and hard to read.

-- The color choice (Ugi green / Makira red all along the manuscript) may be especially challenging for colorblind readers.

-- On figure 1G, confidence intervals are hardly visible on the gray background and invisible on black.

-- The curves in suppl fig S2C-S2F are assignment probabilities of SA/SC ancestry or Ugi ancestry, because these curves were smoothened with gglot2,unfortunately leading to probabilities that may appear negative or >1 on some intervals, which is inconsistent with a probability.

-- In the legends of fig. S2 and S7, I would not call "circle" what is rather a dot or a point.

5- Typos:

-- Legend of fig S8: Tress -> Trees

-- Line 437: and -> an

Reviewer #3: Review is attached

**Have all data underlying the figures and results presented in the manuscript been provided?**

Reviewer #1: Yes

Reviewer #2: **No: **Computer code is available, however it is not clear that the data numerical underlying most graphs is made available as a spreadsheet as recommended in the PLOS genetics policy

Reviewer #3: Yes

PLOS authors have the option to publish the peer review history of their article (what does this mean?). If published, this will include your full peer review and any attached files.

Reviewer #1: No

Reviewer #2: No

Reviewer #3: **Yes: **Yann Bourgeois

---

## [Editor Report · Decision Letter 1]

29 Sep 2022

Dear Dr Campagna,

Thank you very much for submitting your Research Article entitled 'Independent selective sweeps on pigmentation genes mediate convergent evolution of island melanism in two incipient bird species' to PLOS Genetics.

The manuscript was fully evaluated at the editorial level and by independent peer reviewers. The associate editor appreciated the attention to an important topic but identified some concerns that we ask you address in a revised manuscript.

We therefore ask you to modify the manuscript according to the review recommendations. Your revisions should address the specific points made by each reviewer.

Yours sincerely,

Nicolas Bierne

Academic Editor

PLOS Genetics

Kirsten Bomblies

Section Editor

PLOS Genetics

Dear Dr. Campagna and Dr Uy,

Thank you for your work of revision. I think you have accurately addressed referees’ concerns and that your ms is near ready for definitive acceptance. However I would like to give you a last opportunity to improve the clarity of the explanations about the ASIP contig sweep(s). Indeed I am not entirely satisfied with the “in and/or around” argumentation. Either it’s in, or it’s around, but “in and around” is not clear to me. You have a GWAS peak, a differentiation peak, an amino acid change (Ile55Thr), and a shallow genealogy (Fig. S6, S8) at ASIP (position ~6.6Mb), and a second GWAS peak, differentiation peak, soft-sweep like genealogy at the 5’ side of MMP24 (position ~7Mb). The interpretation of these two regions would need to be better articulated, not just mixed in a single blurred “in or around” argumentation. To say it more explicitly, what if the 7Mb region was the soft shoulder of a harder sweep (Schrider et al. 2015 doi.org/10.1534/genetics.115.174912)? How is the age of the Ile55Thr substitution dated? If recombinants are included in the window, the age can be much overestimated. What’s the idea in the consecutive sweeps scenario, that the second one makes the plumage blacker? Please try to rework on clarifying the explanations.

I still think the title is misleading. You’d rather say “selective sweeps on independent/different/alternative pigmentation genes” than “independent selective sweeps on pigmentation genes”.

Best regards,

Nicolas Bierne

---

## [Editor Report · Decision Letter 2]

12 Oct 2022

Dear Dr Campagna,

We are pleased to inform you that your manuscript entitled "Selective sweeps on different pigmentation genes mediate convergent evolution of island melanism in two incipient bird species" has been editorially accepted for publication in PLOS Genetics. Congratulations!

Yours sincerely,

Nicolas Bierne

Academic Editor

PLOS Genetics

Kirsten Bomblies

Section Editor

PLOS Genetics

Comments from the reviewers (if applicable):

Dear Dr. Campagna and Dr Uy,

Thank you for this last work of revision. I’m a happy to now definitively accept your manuscript for publication in PLOS Genetics.

Best regards,

Nicolas Bierne

**Data Deposition**

http://datadryad.org/submit?journalID=pgenetics&manu=PGENETICS-D-22-00667R2

**Press Queries**

---

## [Editor Report · Acceptance letter]

17 Oct 2022

PGENETICS-D-22-00667R2 

Selective sweeps on different pigmentation genes mediate convergent evolution of island melanism in two incipient bird species 

Dear Dr Campagna, 

We are pleased to inform you that your manuscript entitled "Selective sweeps on different pigmentation genes mediate convergent evolution of island melanism in two incipient bird species" has been formally accepted for publication in PLOS Genetics! Your manuscript is now with our production department and you will be notified of the publication date in due course.

With kind regards,

Olena Szabo

PLOS Genetics

On behalf of:
